# Swarm Intelligence Algorithms in Text Document Clustering with Various Benchmarks

**DOI:** 10.3390/s21093196

**Published:** 2021-05-04

**Authors:** Suganya Selvaraj, Eunmi Choi

**Affiliations:** 1Department of Financial Information Security, Kookmin University, Seoul 02707, Korea; suganya@kookmin.ac.kr; 2Department of Software, College of Computer Science, Kookmin University, Seoul 02707, Korea

**Keywords:** swarm intelligence algorithms, text document clustering, artificial intelligence, data mining

## Abstract

Text document clustering refers to the unsupervised classification of textual documents into clusters based on content similarity and can be applied in applications such as search optimization and extracting hidden information from data generated by IoT sensors. Swarm intelligence (SI) algorithms use stochastic and heuristic principles that include simple and unintelligent individuals that follow some simple rules to accomplish very complex tasks. By mapping features of problems to parameters of SI algorithms, SI algorithms can achieve solutions in a flexible, robust, decentralized, and self-organized manner. Compared to traditional clustering algorithms, these solving mechanisms make swarm algorithms suitable for resolving complex document clustering problems. However, each SI algorithm shows a different performance based on its own strengths and weaknesses. In this paper, to find the best performing SI algorithm in text document clustering, we performed a comparative study for the PSO, bat, grey wolf optimization (GWO), and K-means algorithms using six data sets of various sizes, which were created from BBC Sport news and 20 newsgroups. Based on our experimental results, we discuss the features of a document clustering problem with the nature of SI algorithms and conclude that the PSO and GWO SI algorithms are better than K-means, and among those algorithms, the PSO performs best in terms of finding the optimal solution.

## 1. Introduction

Text document clustering is the application of cluster analysis referring to the unsupervised classification of textual documents into clusters based on content similarity. Text document clustering can be applied in organizing large document collections, extracting hidden information from data generated by IoT sensors, finding similar documents, detecting duplicate content, search optimization, and recommendation systems [1,2]. These text document sources may come from web pages, blog posts, news articles, or other text files [3]. Extracting relevant information from the data is a challenging task that needs fast and high-quality document clustering algorithms. The most popular algorithms for clustering are K-means and its variants such as bisecting K-means and K-medoids [4]. The K-means algorithm is a simple, fast, and unsupervised partitioning algorithm offering easily parallelized and intuitive results [5].

Swarm intelligence (SI) is the collective behavior of self-organized and decentralized systems, which includes simple and unintelligent individuals that follow simple rules or behaviors to accomplish very complex tasks with limited local information. SI algorithms have been discovered from observing natural or artificial behaviors such as bird flocks, fish schools, and ant food foraging [6]. SI algorithms include particle swarm optimization (PSO), the bat algorithm (BA), grey wolf optimization (GWO), the firefly algorithm (FFA), ant colony optimization (ACO), the artificial fish swarm algorithm (AFSA), and artificial bee colony optimization (ABC) [7]. These algorithms can be used in many real-world problems such as traveling salesman problems (TSPs), feature selection (FS), robot swarm learning, cluster analysis, and scheduling. SI algorithms have advantages like adaptability, robustness, scalability, and flexibility over traditional clustering algorithms [8]. These characteristics of SI algorithms make them suitable for solving complex text document clustering problems.

However, each SI algorithm shows a different performance based on its own strengths and weaknesses, which necessitates a comparative evaluation of the SI algorithms to find the best one that offers the optimal solution. In this paper, therefore, we perform a comparative study of PSO, BA, GWO, and K-means using six data sets of various sizes which were created from BBC Sport [9] news and 20 newsgroups [10] to find the best performing SI algorithm in text document clustering. Here, the text documents were preprocessed using natural language processing, and different metrics such as purity, homogeneity, completeness, V-measure, ARI, and average running time were used for the comparison [7]. Based on experimental results, we discuss the features of a document clustering problem with the nature of SI algorithms and conclude that PSO and GWO are better than the traditional K-means clustering algorithm and PSO is the best performing algorithm in terms of finding the optimal solution.

## 2. Related Work

This section summarizes the previous comparative study results for SI algorithms in a few applications such as TSP, FS, robot swarm learning, scheduling, clustering, and document clustering.

Shima Sabet and Farokhi [11] implemented the TSP with SI algorithms such as the genetic algorithm (GA), PSO, ABC, and ACO, and the performance of these algorithms was compared. This study states that ABC shows better performance than other algorithms and also ACO shows good performance for less than 80 cities in TSP. A study by Basir and Ahmad [12] compared a few SI algorithms for FS and suggests that SI needs to be studied more for FS. Fan et al. [13] performed a comparative study for PSO, GWO, and BA in robot swarm learning and their result shows that PSO outperforms BA and GWO. In addition, GWO performs better than BA and PSO for a large number of robots and a long communication range. S.J.Mohana [14] reported a comparative analysis study for cloud scheduling for PSO and ACO algorithms and the study shows that PSO performs better than ACO. Elhady and Tawfeek [15] compared ABC, PSO, and ACO in dynamic cloud task scheduling and the results show that ABC outperforms PSO and ACO. Ref. Zhu et al. [16] compared twelve SI algorithms for an uninhabited combat air vehicle path-planning problem and their results show that the spider monkey optimization (SMO) algorithm performs better than others in discovering a safe path.

The SI algorithms FFA, CS, BA, and PSO were compared in medical data clustering and results revealed that CS performs slower than others and PSO and BA are relatively faster than others [17]. Figueiredo et al. [18] studied the comparison of SI algorithms in the clustering domain and concluded that PSO is the best performing algorithm. Algorithms ABC, ACO, and AFSA are the second-level performers. Additionally, SI algorithms such as the whale optimization algorithm (WOA) and BA, and metaheuristic algorithm differential evolution (DE), were combined with the proposed method OpStream for data stream clustering and these results were compared with the state-of-art algorithms DenStream and CluStream [19,20,21]. Lu et al. [22] proposed the text clustering PSO (TCPSO) algorithm by extending the PSO algorithm to solve the variable weighing in text clustering and compared these results with a few other algorithms such as bisection K-means and K-means. Judith and Jayakumari [23] proposed a hybrid algorithm that includes K-means and PSO to solve the distributed document clustering problem. Abualigah et al. [24] compared a few nature-inspired optimization algorithms in text document clustering and the result showed that, according to the accuracy measure and F-measure, GWO is the best performing algorithm and GA is the least performing algorithm. Rashaideh et al. [25] studied the GWO algorithm in a text document clustering problem and their result shows that GWO performs better than the β-hill climbing and hill-climbing algorithms.

## 3. Process of Text Document Clustering

This section describes the main processes of text document clustering such as text document preprocessing, document representation, clustering, and cluster validation.

As shown in Figure 1, preprocessing converts all given documents into a document matrix. Then, algorithms such as K-means and SI algorithms (PSO, GWO, and BA) are used to create text document clusters by using the document matrix. A cluster validation step is used to validate the clusters and performance of the clustering techniques.

### 3.1. Text Document Preprocessing

The preprocessing step greatly influences the result of clustering. Here, we applied a preprocessing step to convert the set of documents into a mathematical data model with which the computer can deal [26,27]. The result of this preprocessing is a mxn term-document matrix. This matrix is built with the table of frequencies and occurrences of terms in each document. Here, *m* is the number of unique terms in the document collection and *n* is the number of documents in the collection. We use the techniques of tokenization, stop word removal, stemming, and term weighting involved in the preprocessing as follows [26].

 Tokenization: Tokenization is a process that splits a stream of text documents into words or terms and removes empty sequences. Here, each word or symbol is taken from the first character to the last character, where each word is called a token [28]. However, sometimes defining a “word” is difficult. A tokenizer often depends on simple heuristics such as [29]:
- the resulting list of tokens may or may not include punctuation and white space;- tokens are separated by white space characters such as spaces, line breaks, or punctuation characters.
 Stop word removal: Stop words such as an, are, for, be, and other common words are more frequent and short functional words also take small weighting. These words should be removed from the document to increase the performance of text document clustering. The list of stop words is available at http://members.unine.ch/jacques.savoy/clef/index.html (accessed on 22 April 2021) and includes 571 words [28]. Stemming: Stemming is the process of reducing inflectional words to the same root by removing the affixes (prefixes and suffixes) of each word. For example [26,29]:
- section, dissect, and intersect all have a common origin or source Sect called the feature;- if the word ends with *ed* or *ing* or *ly*, we remove the *ed* or *ing* or *ly*, respectively.
 Term weighting: The term weighting is assigned for each term or feature by considering the frequency of each term in the document. The term frequency-inverse document frequency (TF-IDF) is widely used in weighting methods. Each document is represented as a vector of term weights as shown in Equation (Equation 1) [28].
(1)$$d_{i}=\left(w_{i,1},w_{i,2},\cdots ,w_{i,t}\right)$$

The term weight for the feature j in document i is calculated using the following Equation (Equation 2).
(2)$$w_{i,j}=tf\left(i,j\right)\times idf\left(i,j\right)=tf\left(i,j\right)\times log\left(\frac{n}{df(j))}\right)$$

In Equation (Equation 2), $w_{i,j}$ represents the weight of document *i* and term *j*. tf(i,j) denotes the occurrences of term *j* in document *i*. idf(i,j) is the inverse document frequency. *n* represents the number of all documents in the data set and df(j) represents the number of documents that contain feature *j*. Usually, the vector space model (VSM) is used in text mining to represent the document features as a vector (row) of weight. Equation (Equation 3) is the common format of the VSM and that includes *n* documents and each document includes *t* terms.
(3)$$VSM=\left[\begin{array}{ccccc} \; & w_{1,1} & \cdots & w_{1,(t-1)} & w_{1,t}\\ \; & \vdots & \ddots & \vdots & \cdots \\ \; & \cdots & \cdots & \cdots & \cdots \\ \; & w_{(n-1),1} & \cdots & \cdots & w_{(n-1),t}\\ \; & w_{n,1} & \cdots & w_{n,(t-1)} & w_{n,t} \end{array}\right]$$

### 3.2. General Process of SI Algorithms in Text Document Clustering

This section describes the general process of SI algorithms in text document clustering [30].

As shown in Figure 1, the text document clustering algorithms directly use the document matrix to convert the set of document data sets into meaningful sub-collections [28]. Some common fundamental phases of SI algorithms are 1. initialize the population of swarm agents; 2. define the termination condition; 3. create document clusters using agents’ positions (centroids); 4. evaluate the fitness function; 5. agent communication; 6. update and move agents; and 7. return the global best solution.

Before the initialization of the algorithm, the values of the algorithm parameters should be defined. Each algorithm has different parameters and those parameters may affect the result of the algorithm. The evolutionary process of the SI algorithm starts with initialization. In the second phase, the termination condition needs to be defined to stop the execution of the algorithm. Here, we use the number of iterations as the stopping criterion. In this experiment, we used different iteration values (1, 10, 20, …, 90, and 100) to evaluate and compare the performance of algorithms with different iterations. In the third phase, we create clusters by assigning each document to the closest center (agent’s position) by calculating the distance to each centroid. The fourth step evaluates the fitness function that is responsible for the evaluation of search agents. The fitness function is either single or combined. Usually, one of the metrics is used as a fitness function. In this study, we use the purity metric as a fitness function. The highest value of purity is considered the best fitness solution. Here, agent communication is used to share information among agents. For example, in the PSO algorithm, agents share the personal best solutions among agents. Agents in an SI algorithm update and move based on some mathematical background of the algorithm. In the last step, after meeting the stopping criterion the SI algorithm returns the best fitness solution from the search agent.

### 3.3. Clustering Evaluation Metrics

Cluster evaluation is a post-clustering technique that validates the quality of final results from clustering algorithms. There are two kinds of validating criteria available: external and internal. External criteria measure the performance by matching clustering structure to a priori knowledge, which includes metrics such as purity, entropy, F-measure, homogeneity, accuracy, completeness, V-measure, and adjusted rand index (ARI). In this study, we used external evaluation metrics such as purity, homogeneity, completeness, V-measure, and ARI to measure the quality of text document clustering algorithms as shown in Figure 1.

#### 3.3.1. Purity

Purity measures whether the clusters contain documents from a single category. The purity value ranges between 0 to 1 and the purity value of an ideal cluster is 1. The purity value is computed by dividing the number of correctly assigned documents by *N* and the formula is defined as follows [31,32]:(4)$$Purity\left(\Omega ,C\right)=\frac{1}{N}\sum_{k}^{}max_{j}\left|\omega_{k}\cap c_{j}\right|$$
where $\Omega =\{\omega_{1},\omega_{2},\cdots ,\omega_{j}\}$ is the set of clusters and $C=\{c_{1},c_{2},\cdots ,c_{j}\}$ is the set of classes.

#### 3.3.2. Homogeneity, Completeness, and V-Measure

Both homogeneity and completeness range between 0 and 1. Homogeneity is 1 if all of its clusters contain only data points that are members of a single class. Completeness is 1 if all the data points that are members of a given class are elements of the same cluster.

Homogeneity (Equation (Equation 5)) and completeness (Equation (Equation 6)) values can be calculated as follows [33,34]: (5)$$h=1-\frac{H(C|K)}{H(C)}$$
(6)$$c=1-\frac{H(K|C)}{H(K)}$$
where H(C|K) is the conditional entropy of the classes defined as:(7)$$H\left(C|K\right)=-\sum_{c=1}^{\left|C\right|}\sum_{k=1}^{\left|K\right|}\frac{n_{c,k}}{n}\times \log\left(\frac{n_{c,k}}{n_{k}}\right)$$
and H(C) is the entropy of the classes defined as: (8)$$H\left(C\right)=-\sum_{c=1}^{\left|C\right|}\frac{n_{c}}{n}\times \log\left(\frac{n_{c}}{n}\right)$$

Here, *n* is the total number of samples. $n_{c}$ and $n_{k}$ are the number of samples belonging to class *c* and cluster *k*, respectively. $n_{c,k}$ is the number of samples from class *c* assigned to cluster *k*.

V-measure can be defined as the harmonic mean of homogeneity and completeness as follows [33,34]:(9)$$v=2\times \frac{h\times c}{h+c}$$

#### 3.3.3. ARI

The rand index (RI) measures the similarity between two data clusters by considering all pairs of samples and counting pairs that are assigned in the same or different clusters in the predicted and true clusters.

The ARI value is adjusted for the chance of raw RI and can be defined as follows [34,35]:(10)ARI=(RI−ExpectedRI)/(max(RI)−ExpectedRI)

The ARI ranges between 0 and 1. The value of ARI is 1 if the clusters are identical.

## 4. Algorithms for Text Document Clustering

This section shows the modified structure of K-means and SI algorithms such as PSO, BA, and GWO to perform document clustering.

### 4.1. K-Means Clustering Algorithm

K-means is a simple and iterative algorithm that clusters a group of data sets into a K predefined number of clusters [36]. Algorithm 1 shows a basic K-means clustering algorithm.

The K-means algorithm starts with random initialization of cluster centers. In step 2, the algorithm assigns each document into its closest centroid based on their similarity. Step 3 recalculates the centroid of each cluster. Steps 2 and 3 will repeat they until meet the termination condition.
**Algorithm 1:****K-means Document Clustering Algorithm**Step 1. Randomly select K points as the initial cluster centers.Step 2. Assign all points (documents) to the closest centroid.Step 3. Recalculate the centroid of each cluster.Step 4. Repeat steps 2 and 3 until termination condition is reached.

### 4.2. SI Algorithms for Document Clustering

#### 4.2.1. PSO Algorithm

The PSO algorithm is an optimization algorithm developed based on bird and fish flock movement behavior. The PSO algorithm aims to locate all the particles in the optimal position in a multi-dimensional space [36]. In this study, each particle in the PSO algorithm represents a cluster centroid of text document clusters. The initialization phase is used to initialize the number of clusters K and a few PSO parameters. After the initialization, new positions (Equation (Equation 11)) and velocities (Equation (Equation 12)) are calculated for each particle. Each position is considered as a centroid and clusters are created by assigning each document to the closest center (particle’s position) by calculating the distance to each centroid. Then, the fitness function evaluates the clusters, and the global best (gbest) and individual best (pbest) solutions are updated until they reach the termination condition criteria as shown in Algorithm 2. As a result, the algorithm finally returns the global best solution [37].
(11)$${}^{V_{id}^{k+1}}=\omega *v_{id}^{k}+c_{1}r_{1}^{k}\left(pbest_{id}^{k}-x_{id}^{k}\right)+c_{2}r_{2}^{k}\left(gbest_{d}^{k}-x_{id}^{k}\right)$$
(12)$$x_{id}^{k+1}=x_{id}^{k}+v_{id}^{k+1}$$

In Equations (11) and (12), $V_{id}^{k}$ and $x_{id}^{k}$ stand for the speed and position of the particle *i* at iteration *k*. $pbest_{id}^{k}$ is the personal best position of *i*. $gbest_{d}^{k}$ is the global best position among all individuals in the d-dimension. ω is the inertia weight that balances the global search and local search abilities. $c_{1}$ and $c_{2}$ are positive constants used to control the speed of the particle. $r_{1}$ and $r_{2}$ are random parameters within [0, 1] [31,36].
**Algorithm 2:** PSO document clustering algorithm
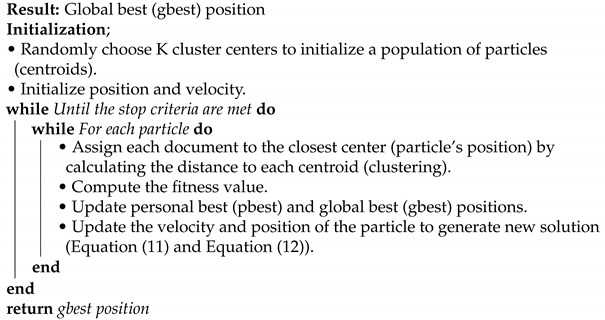


#### 4.2.2. BA Algorithm

The BA algorithm is developed based on the echolocation behavior of microbats. This algorithm follows some idealized rules as follows. 1. All bats use echolocation to sense the distance of prey or predator and also find the barriers in the path. 2. Bats fly with a velocity $v_{i}$ and position $x_{i}$; they have loudness $A_{i}$ and frequency *F* to reach their prey and adjust the frequency of pulse emission *r*. 3. As they get close to the prey, loudness decreases (*A*) and pulse increases (*r*) [6].
(13)$$F_{i}+\left(F_{max}-F_{min}\right)\beta$$
(14)$$V_{i}^{t}=V_{i}^{t-1}+\left(X_{i}^{t}-X^{*}\right)F_{i}$$
(15)$$X_{i}^{t}=X_{i}^{t-1}+V_{i}^{t}$$
(16)$$X_{new}=X_{old}+\varepsilon A^{t}$$
(17)$$r_{i}^{t+1}=r_{i}^{0}\left[1-exp\left(-\gamma t\right)\right],A_{i}^{t+1}$$

In this study, a bat’s position in the BA algorithm represents a cluster centroid of text document clusters. The initialization phase used to initialize the number of clusters K and a few BA parameters is shown in Algorithm 3. After the initialization, Equations (13)–(17) are used to search for the new positions. The centroids and clusters are created by assigning each document to the closest center (particle’s position) by calculating the distance to each centroid. Then, the fitness function evaluates the clusters and the global best (gbest) solution is updated until it reaches the stopping criteria. As a result, the algorithm finally returns the global best solution.

Here, *Q* is the frequency band parameter, β is a random vector in a uniform distribution between 0 and 1. ϵ is a random walk in the normal distribution between −1 and 1. Bats adjust their position according to Equations (13)–(15). Equation (Equation 16) is used to find the local search using a random walk. Equation (Equation 17) is used to find the local search control parameter $r_{i}$ and loudness *A* [38,39].
**Algorithm 3:** BA document clustering algorithm
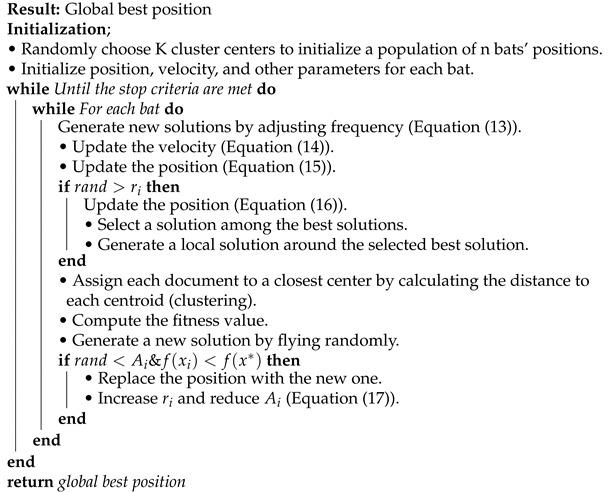


#### 4.2.3. GWO Algorithm

The GWO algorithm mimics the leadership hierarchy and hunting mechanisms of grey wolves in nature. GWO generates three levels of fitness solutions such as alpha (α-first), beta (β-second), and delta (δ-third). The rest of the candidate solutions are assumed to be omega (ω) [6].
(18)$$\vec{X}\left(t+1\right)={\vec{X}}_{p}\left(t\right)+\vec{A}.\vec{D},\vec{D}=\left|\vec{C}.{\vec{X}}_{p}\left(t\right))-\vec{X}\left(t\right)\right|$$
(19)$$\vec{A}=2\vec{A}.\vec{r_{1}}-\vec{a},\vec{C}=2{\vec{r}}_{2},\vec{a}=2-t\frac{2}{MaxIter}$$
(20)$${\vec{D}}_{\alpha }=\left|{\vec{C}}_{1}.{\vec{X}}_{\alpha }-\vec{X}\right|,{\vec{D}}_{\beta }=\left|{\vec{C}}_{2}.{\vec{X}}_{\beta }-\vec{X}\right|,{\vec{D}}_{\delta }=\left|{\vec{C}}_{3}.{\vec{X}}_{\delta }-\vec{X}\right|$$
(21)$${\vec{X}}_{1}=\left|{\vec{X}}_{\alpha }-{\vec{A}}_{1}{\vec{D}}_{\alpha }\right|,{\vec{X}}_{2}=\left|{\vec{X}}_{\beta }-{\vec{A}}_{2}.{\vec{D}}_{\beta }\right|,{\vec{X}}_{3}=\left|{\vec{X}}_{\delta }-{\vec{A}}_{3}.{\vec{D}}_{\delta }\right|$$
(22)$$\vec{X}\left(t+1\right)=\frac{{\vec{X}}_{1}+{\vec{X}}_{2}+{\vec{X}}_{3}}{3}$$

Here, each GWO position in the GWO algorithm represents a cluster centroid of text document clusters. The initialization phase is used to initialize the number of clusters K and a few GWO parameters as shown in Algorithm 4. After the initialization, Equations (18)–(22) are used to search for new positions. The centroid positions and clusters are created by assigning each document to the closest center (particle’s position) by calculating the distance to each centroid. Then, the fitness function evaluates the clusters and the global best (gbest) solution is updated until reaches the stopping criteria. For each iteration, the alpha α-first), beta (β-second), and delta (δ-third) solutions are updated. As a result, the algorithm finally returns the global best solution [40,41].

To calculate $\vec{D}$, $\vec{A}$, and $\vec{C}$ vectors Equation (Equation 18) is used. Equations (20)–(22) are used to update the positions of wolves. Here, $\vec{a}$ is a control parameter, $\vec{A}$ and $\vec{C}$ are coefficient vectors, and $\vec{r_{1}}$ and $\vec{r_{2}}$ are random vectors.
**Algorithm 4:** GWO document clustering algorithm
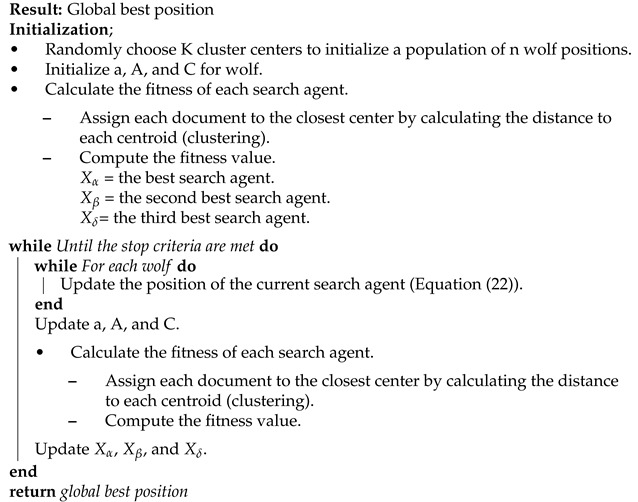


#### 4.2.4. Parameter Set of SI Algorithms

This subsection shows the parameters of all standard SI algorithms and their values. In this study, we focused on comparing the standard SI algorithms. We adopted and used the default parameters of SI algorithms that were introduced in the literature experiments for generic optimization, with different examples [17,42]. Although our experimental problem was text document clustering, which is not an optimization problem, we applied the initial parameter settings of the optimization problem case to ours and tested those standard SI algorithms. We did not use any other parameter fine-tuning methods here.

As shown in Table 1, k represents the number of agents, and its value was 10 in all algorithms for this experiment. In the PSO algorithm parameter values were assigned as ω=0.9, $c_{1}=0.5$, and $c_{2}=0.3$ [42]. In the BA algorithm parameter values were assigned as A=0.5, r=0.5, $Q_{min}=0.0$, and $Q_{max}=2.0$ [17].

#### 4.2.5. Features of SI Algorithms

This section shows the features of SI algorithms, the details of important control and local search parameters, and the best performing area of SI algorithms PSO, GWO, and BA as shown in Table 2.

The text document clustering problems are NP problems. Since there is no deterministic algorithm to solve this problem in a polynomial time, we applied an approximate algorithm to find the nearly optimal solution. The SI algorithms were used to solve the problem in a stochastic manner. Although we could not find the feasibility of the text document clustering in a polynomial time, we could find an approximate feasible solution by using the SI algorithms with 10 simulations for each iteration (1, 10, 20, …, 100) as mentioned in the experiment section.

In PSO, ω, $c_{1}$, and $c_{2}$ are the important control parameters. $r_{1}$ and $r_{2}$ are the local search parameters, which are drawn from uniform distribution [0,1]. PSO is a simple, fast computing, and parallel processing algorithm [43]. Comparing other standard SI algorithms, the standard PSO algorithm performs well in clustering and scheduling applications [14,17,18].

The GWO algorithm includes two main control parameters, $\vec{a}$ and *C*, and local search random vector parameters $r_{1}$ and $r_{2}$ from [0,1]. GWO has a few important features such as faster convergence due to continuous reduction of the search space and fewer decision variables (i.e., α, β, and δ), avoiding local minima, better stability, and robustness [44]. Comparing other standard SI algorithms, GWO performs well in robot swarm learning [13].

The BA has control parameters such as loudness (*A*), pulse rate (*r*), pulse rate adaptation (γ), loudness decrements (α), and local search radius and frequency band parameters (*Q*), and also has local search parameters such as the random vector (β) from a uniform distribution [0,1] and a random walk ϵ from the normal distribution [−1,1]. The BA provides very quick convergence at the initial stage by switching from exploration to exploitation [45]. Comparing other standard SI algorithms, the BA also performs well in clustering [17].

## 5. Experiment

The experimentation shows benchmark data sets, experimental conditions, and metrics that were used for the experimentation.

### 5.1. Benchmark Data Sets

In an unsupervised clustering context, classes and class memberships are unknown, and these must be discovered by the clustering algorithm. The performance of these algorithms can be determined by using data sets with known clusters for benchmarking in unsupervised clustering [46]. In this study, we used six different benchmark data sets with different numbers of documents, terms (features), and clusters which were created from BBC Sport news and 20 newsgroup data sets as shown in Table 3. The BBC Sport news data set is from BBC News and is provided to use as a machine learning benchmark consisting of 737 text documents including 4613 terms under five classes: athletics, rugby, cricket, tennis, and football [9]. The 20 newsgroups are popular machine learning data sets for text applications such as text clustering that consist of 20,000 messages taken from 20 different newsgroups such as misc.forsale, soc.religion.christian, and so on [10].

As shown in Table 3, Data set 1 includes 1427 documents and 23,057 terms and these documents belong to 2 clusters from 20 newsgroups. Data set 2 consists of 737 documents and 4613 terms and belongs to 5 clusters from BBC Sport news. Data set 3 includes 40 documents and 2596 terms and belongs to 5 clusters from BBC Sport news. Data set 4 includes 200 documents and 8716 terms and these documents belong to 4 clusters from 20 newsgroups. Data set 5 includes 100 documents and 5549 terms and these documents belong to 3 clusters from 20 newsgroups. Data set 6 includes 100 documents and 3876 terms belonging to 2 clusters from BBC Sport news.

### 5.2. Experimental Conditions

In this study, Ubuntu OS, Python3, and Jupyter Notebook were used for the experiments. The SI algorithms PSO, BA, and GWO are stochastic. Hence, for the same iteration number different best solutions can be obtained. Therefore, 10 simulations were used for each iteration number, and the best fitness was taken among them for each algorithm.

### 5.3. Results

In this study, we used standard PSO, BA, and GWO for the text document clustering with six data sets, and the performance of these algorithms was evaluated using various metrics such as purity, homogeneity, completeness, V-measure, ARI, and average running time. These results were compared with the K-means clustering algorithm.

Table 4 shows the mean and standard deviation results for all iteration numbers and their performance ranks for all clustering algorithms with six data sets from BBC news and 20 newsgroups. The performance of each algorithm (K-means, PSO, GWO, and BA) was measured using metrics such as purity, homogeneity, completeness, V-measure, and ARI, and those results are displayed in Table 4. The highest purity, homogeneity, completeness, V-measure, and ARI values of each algorithm for the data sets are highlighted in bold. The rank is displayed for each algorithm based on the average evaluation metric values for each and all data sets. Here, the PSO algorithm has the highest purity mean values (0.724, 0.873, 0.689, 0.848, and 0.998) and BA has the lowest purity mean values (0.636, 0.786, 0.607, 0.728, and 0.976) for data sets 1, 3, 4, 5, and 6, respectively. As shown in Table 4, in terms of all metrics, K-means, PSO, GWO, and BA have ranks 3, 1, 2, and 4, respectively, for data sets 1, 3, 4, 5, and 6. K-means achieved the highest purity mean value of 0.929 and BA achieved the lowest purity mean value of 0.692 for data set 2. The other metrics’ mean values also provide the same data pattern and rank as purity for all data sets and algorithms as shown in Table 4.

In terms of all evaluation metrics (purity, homogeneity, completeness, V-measure, and ARI), the total performance ranks for algorithms K-means, PSO, GWO, and BA are 3, 1, 2, and 4, respectively. Comparing all results, PSO performs best except for data set 2. Based on the rank, PSO shows the best performance, and the second-best is GWO. The third is K-means and BA performs last.

Figure 2, Figure 3, Figure 4, Figure 5, Figure 6 and Figure 7 show the comparative results for SI algorithms and K-means with different data sets 1, 2, 3, 4, 5, and 6 using a purity metric.

Figure 2 shows the comparison of the algorithms based on purity with data set 1. This result clearly shows that PSO has the best performance over others and the second best is GWO. K-means and BA have third and fourth positions, respectively.

Figure 3 shows the comparison of the algorithms with data set 2. This result clearly shows that K-means has the best performance over the others and the second best is PSO. GWO and BA have third and fourth positions, respectively. Only for iteration number 1 was less purity observed and for the remaining iterations, stable purity is observed.

Figure 4 shows the algorithm comparison with data set 3. It shows the PSO, GWO, and BA perform first, second, and third, respectively. K-means performs lower than all other SI algorithms.

Figure 5 and Figure 6 show a comparison of the algorithms based on data sets 4 and 5, respectively. Here, based on purity results, almost the same results are shown that PSO and GWO perform first and second, respectively. BA and K-means show almost equal and lower performance than PSO and GWO.

Figure 7 shows the algorithm comparison based on data set 6. Here, PSO, GWO, BA, and K-means show almost equal good performance in all iterations.

As shown in Figure 8, the average of the total running time of SI algorithms in each data set was used to evaluate the computational efficiency of the algorithms. Here, K-means is much faster than PSO, GWO, and BA with shorter average running time which is less than a second. Here, PSO takes 17.92, 4.83, 1.83, 30.6, 7.8, and 4.21 average running times in seconds for executing data sets 1, 2, 3, 4, 5, and 6, respectively. This is comparatively more execution time than other algorithms. GWO and BA have an almost equal average running time for all data sets and are faster than the PSO algorithm. Although the PSO has a longer execution time, other evaluation metrics show that the performance of PSO is better at finding the optimum solution.

### 5.4. Discussion

Based on the above results, on average PSO performs best compared to GWO, BA, and K-means. The GWO algorithm performs well compared to BA and K-means. The results show that almost all algorithms provide stabilized results only after the 20th iteration and above. The results show that the algorithms achieve lower maximum purity (0.75) for a larger number of terms (data sets 1 and 4) and higher maximum purity values (close to 1) for a small number of terms (data sets 2, 3, 5, and 6). However, in terms of computational efficiency, K-means is much faster than other algorithms. Although the average running time of PSO for almost all data sets is relatively (approximately 50%) higher than other algorithms, PSO offers the best performance to find the optimal solution.

Among a number of SI algorithms for solving document clustering, PSO finds the optimal solution compared to other SI and K-means traditional algorithms. The PSO algorithm explores efficiently in search space and exploits the best solutions found by tracking global best and local best solutions. These characteristics help the PSO to solve the document clustering problem efficiently. The GWO algorithm is good at exploring new areas. The document clustering problem is not suitable for the exploring expert. The BA algorithm is good at governing and dominating the prey. The document clustering problem, however, has a whole set of documents together. Applying a velocity vector in the BA does not change the essential grouping of document similarities. That is why the nature of the BA algorithm is not fit for the document clustering problem.

The nature and characteristics of SI algorithms are suitable to apply in many domains such as feature selection, finding an optimal route, job scheduling, role-based learning, and clustering. However, variants of SI algorithms such as improved and hybridized SI algorithms are performing well in many optimization tasks compared to the standard SI algorithms [6]. Based on proper parameter setting and tuning schemes, the SI algorithms can be applied to the appropriate domain of problem-solving [7].

## 6. Conclusions

The specific characteristics of each SI algorithm are suitable for solving specific optimization problems such as feature selection, finding an optimal route, job scheduling, role-based learning, and clustering. To handle specific optimization tasks, finding the proper SI algorithm is important. In this study, we compared a few important SI algorithms (PSO, BA, and GWO) for the text document clustering problem. These results were also compared with the K-means traditional clustering algorithm. To compare the performance of these algorithms, a few metrics (purity, homogeneity, completeness, V-measure, and ARI) were used with the news articles of six benchmark data sets of various sizes. The comparison of these results shows that the PSO performs best compared to GWO, BA, and K-means. The GWO algorithm performs well compared to the BA and K-means. To compare the computational efficiency, the average running time was used and the results show that compared to other algorithms, K-means takes far less time and PSO takes more time for execution. PSO achieved better results in terms of solving problems but took more time for execution, which demands more study to improve the algorithms in different aspects. As future work, we are aiming to optimize the SI algorithms for better results in the document clustering domain.

## Figures and Tables

**Figure 1 sensors-21-03196-f001:**
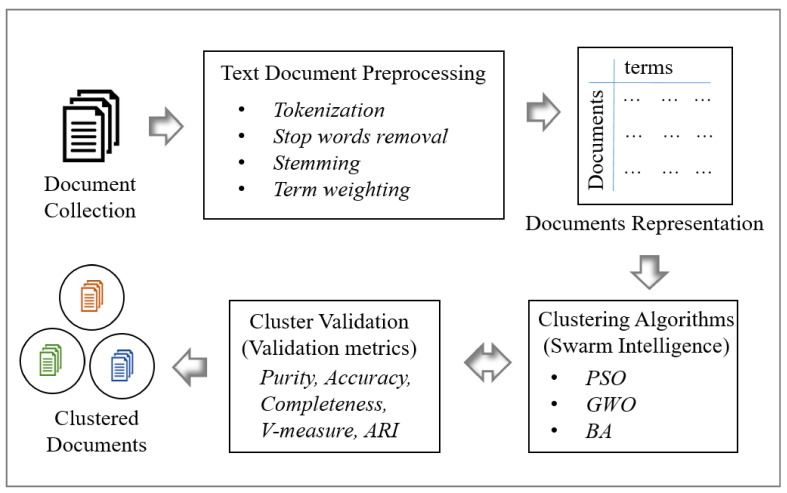
Process of text document clustering.

**Figure 2 sensors-21-03196-f002:**
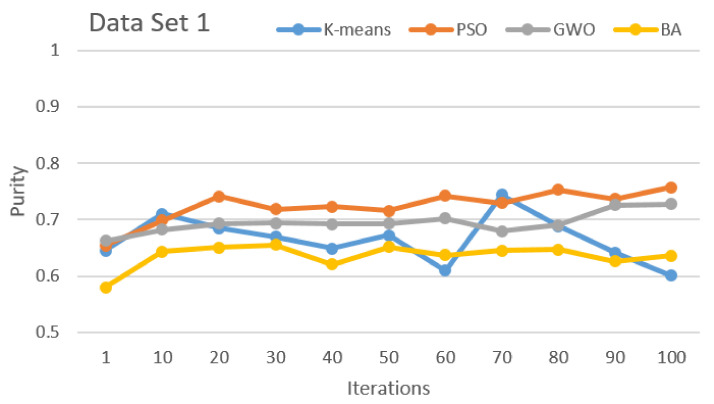
Purity comparison for data set 1.

**Figure 3 sensors-21-03196-f003:**
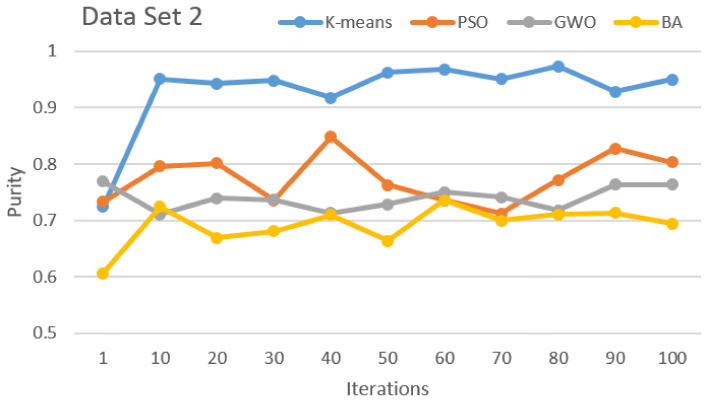
Purity comparison for data set 2.

**Figure 4 sensors-21-03196-f004:**
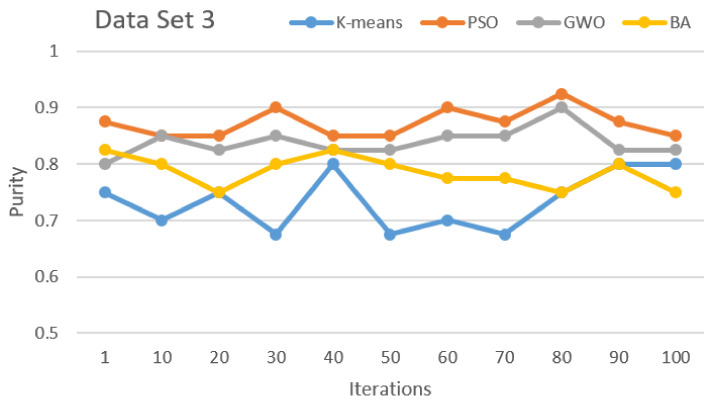
Purity comparison for data set 3.

**Figure 5 sensors-21-03196-f005:**
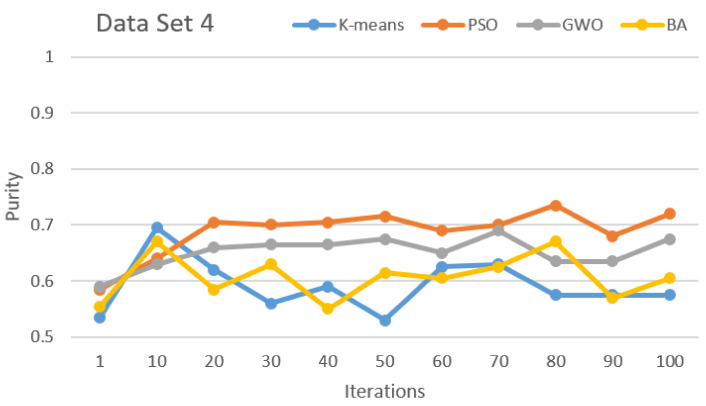
Purity comparison for data set 4.

**Figure 6 sensors-21-03196-f006:**
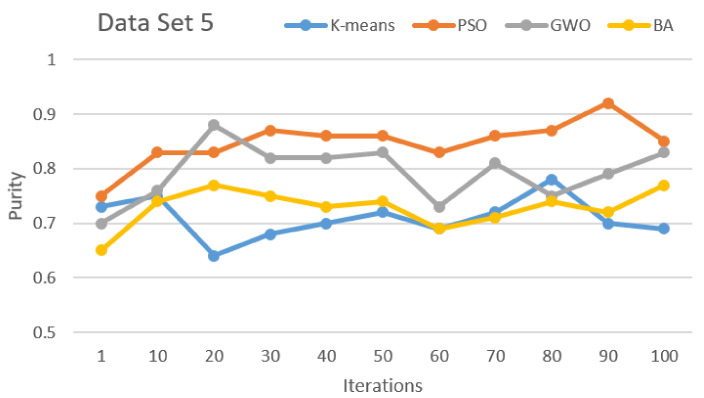
Purity comparison for data set 5.

**Figure 7 sensors-21-03196-f007:**
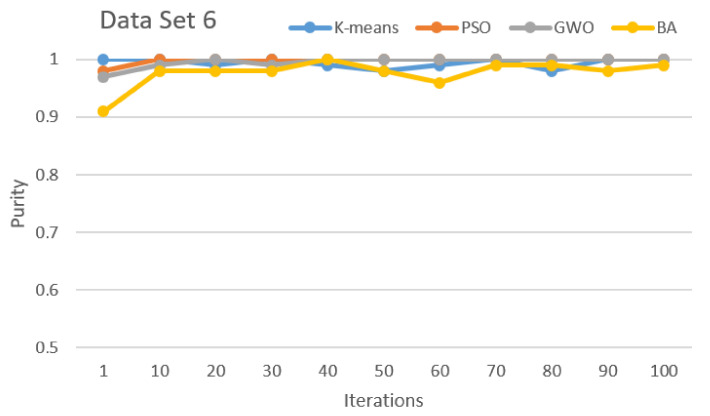
Purity comparison for data set 6.

**Figure 8 sensors-21-03196-f008:**
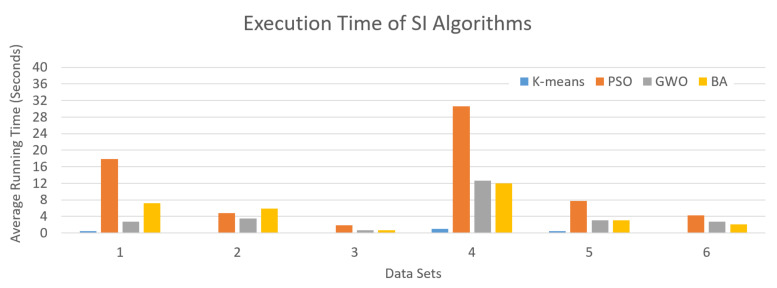
Average running time of SI algorithms.

**Table 1 sensors-21-03196-t001:** Parameter set for SI algorithms.

PSO	Values	BA	Values	BA	Values
k	10	k	10	k	10
ω	0.9	A	0.5		
C1	0.5	r	0.5		
C2	0.3	Qmin	0.0		
		Qmax	2.0		

**Table 2 sensors-21-03196-t002:** Parameters, features, and best performing areas of SI algorithms.

Algorithms	Parameters	Features	Best Performing Area
PSO	Control: ω, $c_{1}$, and $c_{2}$; Local Search: $r_{1}$, and $r_{2}$ are drawn from uniform distribution [0,1].	Simple, fast computing speed, and parallel processing [43].	Clustering and scheduling [14,17,18].
GWO	Control: $\vec{a}$ and *C*; Local Search: $r_{1}$, and $r_{2}$ are random vectors [0,1].	Faster convergence due to continuous reduction of search space and fewer decision variables (i.e., α, β, and δ); avoiding local minima; better stability and robustness [44].	Robot swarm learning [13].
BA	Control: loudness (*A*), pulse rate (*r*), pulse rate adaptation (γ), loudness decrements (α), local search radius, and frequency band parameters (*Q*); Local Search: β is random vector uniform distribution [0,1], ϵ is random walk normal distribution [−1,1].	Provides very quick convergence at a very initial stage by switching from exploration to exploitation [45].	Clustering and feature selection [6,17].

**Table 3 sensors-21-03196-t003:** Benchmark data sets.

Data Set	Source	No. of Documents	No. of Terms	No. of Clusters
1	20 newsgroups	1427	23,057	2
2	BBC Sport	737	4613	5
3	BBC Sport	40	2596	5
4	20 newsgroups	200	8716	4
5	20 newsgroups	100	5549	3
6	BBC Sport	100	3876	2

**Table 4 sensors-21-03196-t004:** Mean and standard deviation results for all iteration numbers for clustering algorithms with six data sets.

Data Sets		Metrics	K-Means	PSO	GWO	BA
			Mean (Std.)			
1		Purity	0.665 (0.04)	**0.724 (0.028)**	0.695 (0.018)	0.636 (0.02)
		Homogeneity	0.081 (0.041)	**0.149 (0.036)**	0.108 (0.023)	0.052 (0.014)
		Completeness	0.085 (0.39)	**0.153 (0.039)**	0.112 (0.023)	0.057 (0.016)
		V-measure	0.083 (0.04)	**0.151 (0.037)**	0.110 (0.023)	0.054 (0.014)
		ARI	0.112 (0.057)	**0.203 (0.045)**	0.156 (0.029)	0.0732 (0.019)
	Rank		3	1	2	4
2		Purity	**0.929 (0.066)**	0.775 (0.042)	0.74 (0.02)	0.692 (0.034)
		Homogeneity	**0.833 (0.097)**	0.525 (0.067)	0.468 (0.041)	0.408 (0.058)
		Completeness	**0.854 (0.076)**	0.527 (0.068)	0.472 (0.05))	0.413 (0.059)
		V-measure	**0.843 (0.088)**	0.525 (0.066)	0.470 (0.045)	0.410 (0.058)
		ARI	**0.832 (0.118)**	0.536 (0.089)	0.462 (0.051)	0.393 (0.0665)
	Rank		1	2	3	4
3		Purity	0.734 (0.049)	**0.873 (0.025)**	0.839 (0.025)	0.786 (0.027)
		Homogeneity	0.591 (0.082)	**0.793 (0.059)**	0.733 (0.039)	0.637 (0.046)
		Completeness	0.605 (0.082)	**0.797 (0.058)**	0.735 (0.041)	0.65 (0.046)
		V-measure	0.598 (0.081)	**0.795 (0.058)**	0.734 (0.04)	0.643 (0.045)
		ARI	0.478 (0.087)	**0.727 (0.058)**	0.659 (0.067)	0.552 (0.07)
	Rank		3	1	2	4
4		Purity	0.592 (0.046)	**0.689 (0.04)**	0.652 (0.027)	0.607 (0.039)
		Homogeneity	0.271 (0.059)	**0.36 (0.056)**	0.322 (0.033)	0.284 (0.046)
		Completeness	0.284 (0.062)	**0.37 (0.058)**	0.336 (0.036)	0.301 (0.051)
		V-measure	0.278 (0.06)	**0.365 (0.057)**	0.329 (0.034)	0.292 (0.048)
		ARI	0.248 (0.058)	**0.363 (0.057)**	0.318 (0.029)	0.263 (0.059)
	Rank		3	1	2	4
5		Purity	0.709 (0.036)	**0.848 (0.04)**	0.793 (0.05)	0.728 (0.034)
		Homogeneity	0.309 (0.052)	**0.541 (0.098)**	0.427 (0.075)	0.314 (0.049)
		Completeness	0.323 (0.063)	**0.556 (0.084)**	0.448 (0.082)	0.347 (0.076)
		V-measure	0.315 (0.540)	**0.548 (0.091)**	0.437 (0.078)	0.329 (0.059)
		ARI	0.321 (0.069)	**0.598 (0.080)**	0.473 (0.117)	0.344 (0.066)
	Rank	3	1	2	4
6		Purity	0.994 (0.008)	**0.998 (0.006)**	0.995 (0.009)	0.976 (0.023)
		Homogeneity	0.958 (0.049)	**0.987 (0.041)**	0.45 (0.491)	0.864 (0.093)
		Completeness	0.959 (0.048)	**0.987 (0.04)**	0.972 (0.051)	0.867 (0.089)
		V-measure	0.958 (0.049)	**0.987 (0.04)**	0.972 (0.051)	0.866 (0.091)
		ARI	0.975 (0.031)	**0.992 (0.027)**	0.982 (0.035)	0.909 (0.085)
	Rank		3	1	2	4
	Total rank		3	1	2	4

## Data Availability

This study used data sets from the publicly archived datasets ucd.ie and ucd.edu, and source code from ju.edu.jo (accessed on 22 April 2021).

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
