# Peer review of "Swarm Intelligence Algorithms in Text Document Clustering with Various Benchmarks"

_sensors, 2021, doi:10.3390/s21093196_

Round 1
Reviewer 1 Report
(1) The Abstract section needs to be improved to elaborate the main conclusions and findings of the study, rather than just what the authors have done.
(2) The computational efficiency of an algorithm is also an important index to evaluate the performance of the algorithm. In the comparative analysis of SI algorithms and K-means clustering algorithm, the computational efficiency of different algorithms should be presented and compared in an appropriate way.
(3) The performance of heuristic algorithm greatly depends on the setting of algorithm parameters. Generally, it takes many experiments to determine the better parameter values of the algorithm, rather than simply setting the value of each parameter directly. The authors should supplement and demonstrate the basis of parameter setting in different algorithms. Otherwise, it's far-fetched to say that an algorithm is better.
Reviewer 2 Report
This article needs amends in terms of
- noverlty: three common optimisation algorithms are sued in conjuction with classic clustering... this looks like an academic exercsie. Why is this study important for the scientific community? This should have been commented further and justified.
- literature (the recent literature in optimisation for classification is not mentioned, see e.g. 10.3390/math7121229 , 10.1109/ACCESS.2020.3016784 and 10.1016/j.inffus.2020.06.009)
- rigour:
- what are the correction strategies used for dealing with solution generated by the heuristic algorithms outside the search space? This has to be reported as it has an impact on the performance (see literature on infeasibility and structural bias).
- how were the paramter of the optimisation algorhtms selcted? were they tuned? This is also requirewd and it has to be commented given its imoprtance in terms fo performances of the algorithms.
Finally, this article is more suitable for other MDPI journals as Algorithms or Mathematics (Computational section) but it is completely out of scope in the Sensors journal. For this reason, I do not recommend it for publication here.
Reviewer 3 Report
I suggest that the authors in the introduction section (2nd paragraph) cite one of the latest reviews on the SI topic: https://www.mdpi.com/2076-3417/8/9/1521. In the first part of the paper, all SI algorithms are presented and discussed.
In the introduction section, the authors must present the motivation behind this paper. The introduction section is also missing a clearer problem statement - point out why this problem is worth researching. Furthermore, point out the main contributions of the paper.
The Related work section should be extended with more document clustering works.
Preprocessing step should be presented more in-depth in the paper. The authors could also add some examples.
I suggest that the authors add to the presented algorithms also references to the equations presented in the text. For example, Algorithm 2, step 5: Update velocity and position of each particle (eq. 11, eq. 12).
How were the parameters set for each algorithm? Was some parameter tuning utilized?
I suggest that section 5 is renamed to Experiment.
I suggest that the authors expand this paper with the Discussion section. Inclusion of the statistical analysis would also be a big plus.
Used references are relevant and up-to-date.
The authors should also prepare at least the following sections at the end: Author Contributions, Funding, Data Availability Statement, and Acknowledgments.
Round 2
Reviewer 1 Report
In view of my concerns, the authors have made revisions and improvements. I recommend accepting this paper, although I am not entirely satisfied with the response to the setting of the algorithm parameters. In addition, in Line #376, There is a grammatical error in the sentence "the performance PSO is better".
Author Response
Dear Editor,
We would like to thank the reviewers and editor for spending their valuable time in reviewing our paper entitled “Swarm Intelligence Algorithms in Text Document Clustering with Various Benchmarks” (Manuscript ID: sensors-1185560). We appreciate the reviewers’ continuous interest in making our manuscript better. The comments from the reviewers and editor are grateful and helped us to improve the quality and readership of our manuscript. Here, we revised our manuscript based on the comments from the reviewers. We hope the revised version is now suitable for publication in the journal ‘Sensors’. We attached our responses to the Reviewer’s comments altogether.
Please note that the reviewer comments are highlighted with ‘bold text in dark red color’ and our responses are given in ‘normal text in black fonts’. The modifications made in the text are highlighted as ‘italicized black fonts’.
Reviewer 1
In view of my concerns, the authors have made revisions and improvements. I recommend accepting this paper, although I am not entirely satisfied with the response to the setting of the algorithm parameters. In addition, in Line #376, There is a grammatical error in the sentence "the performance PSO is better".
Response: We thank the reviewer for the valuable comments and for accepting this paper. In this study, we focus on comparing the standard SI algorithms to observe and compare the performances of algorithms in text document clustering. To work with the standard SI algorithms, we adopt and use default parameters that are introduced in the literature experiments. Those literature experiments used the SI algorithms for the generic optimization, respectively in the References: #17 and #42, with different examples. Although our experimental problem is text document clustering which not an optimization problem, we apply the initial parameter setting of the optimization problem case to ours and test those standard SI algorithms. We did not use any other parameter fine-tuning methods.
The grammatical error in #376 (changed to #387) was corrected properly.
Changes in Manuscript:
Page #9, Line #269~275 changed as follows,
“This subsection shows the parameters of all standard SI algorithms and their values. In this study, we focus on comparing the standard SI algorithms. We adopt and use the default parameters of SI algorithms that are introduced in literature experiments for the generic optimization, with different examples [17] [42]. Although our experimental problem is text document clustering which is not an optimization problem, we apply the initial parameter setting of the optimization problem case to ours and test those standard SI algorithms. We did not use any other parameter fine-tuning methods here.”

Reviewer 2 Report
The authors have clarified all the point raised, but I still believe that not using corrections for dealing with feasible solutions is wrong. The authors claim they will add this check in the future but still, I believe they should mention in the text why they believe that provided solution by the algorithm is feasible (is there any rationale?) or ad a line to explain that solutions have been used in the system only if feasible. It is known that heuristics can easily generate points outside the domain. Hence, this deserves some attention.
Author Response
Dear Editor,
We would like to thank the reviewers and editor for spending their valuable time in reviewing our paper entitled “Swarm Intelligence Algorithms in Text Document Clustering with Various Benchmarks” (Manuscript ID: sensors-1185560). We appreciate the reviewers’ continuous interest in making our manuscript better. The comments from the reviewers and editor are grateful and helped us to improve the quality and readership of our manuscript. Here, we revised our manuscript based on the comments from the reviewers. We hope the revised version is now suitable for publication in the journal ‘Sensors’. We attached our responses to the Reviewer’s comments altogether.
Please note that the reviewer comments are highlighted with ‘bold text in dark red color’ and our responses are given in ‘normal text in black fonts’. The modifications made in the text are highlighted as ‘italicized black fonts’.
Reviewer 2
The authors have clarified all the point raised, but I still believe that not using corrections for dealing with feasible solutions is wrong. The authors claim they will add this check in the future but still, I believe they should mention in the text why they believe that provided solution by the algorithm is feasible (is there any rationale?) or ad a line to explain that solutions have been used in the system only if feasible. It is known that heuristics can easily generate points outside the domain. Hence, this deserves some attention.
Response: We thank the reviewer for this gentle mention. In this paper, we used the benchmark data sets [References: #9 and #10] for text document clustering which includes raw documents and cluster solutions together [Page: #10; Line: #314~316]. To determine the performance of the SI algorithms, we used different metrics such as purity and homogeneity to compare the results of our algorithms with the provided solutions from the benchmark data sets.
These kinds of text document clustering problems are NP problems. Since there is no deterministic algorithm to solve this problem in a polynomial time, we apply an approximate algorithm to find the nearly optimal solution. The SI algorithms are used to solve the problem in a stochastic manner. Although we cannot find the feasibility of the text document clustering in a polynomial time, we could find an approximate feasible solution by using the SI algorithms, as in the experiment of 10 simulations for each iteration (1, 10, 20, …,100) [Page: #12, Line: #334~#336].
Changes in Manuscript:
Page #10, Line #284~290 changed as follows,
“The text document clustering problems are NP problems. Since there is no deterministic algorithm to solve this problem in a polynomial time, we apply an approximate algorithm to find the nearly optimal solution. The SI algorithms are used to solve the problem in a stochastic manner. Although we cannot find the feasibility of the text document clustering in a polynomial time, we could find an approximate feasible solution by using the SI algorithms with 10 simulations for each iteration (1, 10, 20, …,100) as mentioned in the experiment section.”

Reviewer 3 Report
After addressing comments, the manuscript has improved from the previous version and, in my opinion, can be accepted.
Author Response
Dear Editor,
We would like to thank the reviewers and editor for spending their valuable time in reviewing our paper entitled “Swarm Intelligence Algorithms in Text Document Clustering with Various Benchmarks” (Manuscript ID: sensors-1185560). We appreciate the reviewers’ continuous interest in making our manuscript better. The comments from the reviewers and editor are grateful and helped us to improve the quality and readership of our manuscript. Here, we revised our manuscript based on the comments from the reviewers. We hope the revised version is now suitable for publication in the journal ‘Sensors’. We attached our responses to the Reviewer’s comments altogether.
Please note that the reviewer comments are highlighted with ‘bold text in dark red color’ and our responses are given in ‘normal text in black fonts’. The modifications made in the text are highlighted as ‘italicized black fonts’.
Reviewer 3
After addressing comments, the manuscript has improved from the previous version and, in my opinion, can be accepted.
Response: We thank the reviewer for accepting this paper. Your former comments are so valuable to improve our manuscript a lot. We appreciate it so much.
